# OpenReview forum: "Dynamically Scaled Activation Steering"
_TMLR — Under review for TMLR_

### Review · Reviewer_SERQ · 2026-07-04

**Summary Of Contributions:**

This paper generally has five contributions:
1. The authors introduce the idea of adaptive steering strength for steering in the context of generative modeling. The authors propose DSAS, a framework to dynamically scale intervention strength based on activation characteristics. In this framework, the steering strength is not predefined by the users, but generally learned by the model itself.
2. The authors further extend DSAS to end-to-end frameworks like LinEAS, enabling joint training via backpropagation.
3. The authors show that DSAS can outperform previous baselines in terms of the steer-quality Pareto frontier.

**Additional Comments:**

No, I do not have any additional comments.

**Audience:**

Yes

**Audience Explanation:**

This paper proposes a properly motivated method targeting one of the major challenges in modern machine learning. The algorithm is simple, effective and computational cheap. The claims are supported by comprehensive experimental evaluations. In addition, the writing is generally clean and easy to follow. Overall, I find this paper interesting to people in the machine learning community.

**Broader Impact Concerns:**

This paper proposes an effective steering method. In the paper, the authors conducted experiments on toxicity mitigation. However, it will be super easy to reverse the labels and use this method to generate toxic outputs. I believe the authors should invest more in this topic to avoid harmful abuse of DSAS.

**Claims And Evidence:**

Yes

**Claims Explanation:**

1. The DSAS approach uses a light-weight neural network to learn the steering strength. This is consistent with the core philosophy of this paper.
2. The extension to end-to-end steering frameworks like LinEAS is neat and effective. Overall, it is reliable to claim that the authors have successfully extended DSAS to the end-to-end steering context.
3. The claim of state-of-the-art performance is supported by both a thorough Pareto frontier evaluation and case studies in two different modalities, overall convincing.

**Requested Changes:**

1. Despite the thorough and convincing evaluation, the authors only conducted experiments on a super small-scale dataset from the modern machine learning perspective. It is dubious whether DSAS can work at a larger scale. I'd suggest the authors add a larger-scale experiment on some tasks, such as general instruction following or math and code reasoning to prove the real-world application potential of DSAS.
2. There are no ablation studies included in this paper. The number 32 seems pretty ad hoc as well. I'd like the authors to add an ablation on the dataset size. I am pretty curious if the best performance of DSAS over other baselines will be achieved consistently or only at this specific data scale.
3. Can the authors explain if the baselines use the Control Set as well? If not, can the author justify that the improvement comes from the algorithm but not from more data seen during training?

---

> ### Author Response · Authors · 2026-07-14
> **Rebuttal to Reviewer SERQ**
>
> We thank the reviewer for the constructive feedback. We address reviewer's concerns below:
>
> **W1 & W2.**
> The low-data regime is not an incidental choice, it is the regime in which activation steering is the right tool at all. Steering exists precisely because it conditions behavior from a handful of unpaired examples, without weight updates or preference data. Once one has enough data and compute to justify parameter fine-tuning, that is the setting where fine-tuning (e.g. LoRA) becomes the natural choice and steering is no longer the interesting question (Rodríguez et al. 2025). Evaluating DSAS with 32 examples is therefore not a small-scale convenience, it is the operating point steering methods are actually interesting to be deployed in (Li et al, 2024, Rodríguez et al. 2025, Suau et al. 2024).
>
> Appendix D makes this ablation on the dataset size explicit. As the sample count grows from 4 to 512, the discriminator's accuracy rises and its variance falls monotonically. This means the mechanism DSAS depends on only becomes more reliable as data increases. We report in our main manuscript experiments the difficult end of that curve, which is precisely the interesting operating point for activation steering. We have added now a clarification in Limitations and Discussion section.
>
> **W3.**
> > Can the authors explain if the baselines use the Control Set as well?
>
> No baseline uses the control set. All baselines were originally designed to map all inputs to a target space and thus only require a source and a target set of samples to be trained. The control set defines a new region that should never be transported, and DSAS uses it to improve the performance of the baselines.
>
> > If not, can the author justify that the improvement comes from the algorithm but not from more data seen during training?
>
> The improvement cannot come from extra data. More data can only sharpen a steering direction but it can never make an unconditional intervention (CAA, ITI, LinEAS) selective. DSAS operates on top of the base steering and is orthogonal to it. A more accurate direction (whether from more data or a better estimator) makes DSAS more effective, not redundant. The control set is not extra training signal for the direction, it is what tells DSAS when to steer.
>
> Crucially the control set may be completely different from the target dataset, as demonstrated by the blurring experiment in the paper. In the toxicity experiment, we employ 32 additional data samples to make the distinction between the control and target sets explicit. However, these samples are drawn from the same distribution and, in fact, they could even be identical, as discussed in Section 3.2.
>
> To show this empirically, we re-ran CAA+DSAS on Gemma‑2‑2B with the control set replaced by the target set ($\mathcal{C}=\mathcal{T}$, i.e., no extra data), sweeping the global strength $\lambda$. The trade-off is unchanged, confirming the gain is algorithmic, not from the 32 extra samples:
>
> | $\lambda$ | $\text{Tox}_{\text{TET}}$ (distinct $\mathcal{C}$) | $\text{Tox}_{\text{TET}}$ ($\mathcal{C}=\mathcal{T}$) | $\text{PPL}_{\text{Wik}}$ (distinct $\mathcal{C}$) | $\text{PPL}_{\text{Wik}}$ ($\mathcal{C}=\mathcal{T}$) |
> |---------:|:--------------------------------------------------:|:-----------------------------------------------------:|:--------------------------------------------------:|:-----------------------------------------------------:|
> | 0.5 | 12.52 | 12.52 | 14.70 | 14.70 |
> | 1.0 | 9.02 | 8.52 | 14.64 | 14.65 |
> | 1.5 | 6.69 | 5.65 | 14.63 | 14.69 |
> | 2.0 | 5.26 | 5.28 | 14.69 | 14.85 |
> | 2.5 | 3.48 | 4.67 | 14.79 | 14.73 |
> | 3.0 | 2.11 | 2.26 | 14.96 | 15.18 |
> | 3.5 | 1.20 | 1.12 | 15.21 | 15.85 |
> | 4.0 | 0.51 | 1.79 | 15.55 | 15.49 |
> | 4.5 | 0.45 | 0.35 | 16.00 | 17.10 |
> | 5.0 | 0.16 | 0.71 | 16.61 | 16.45 |
>
> The two Pareto fronts coincide within single-seed noise (Gemma‑2‑2B; same behavior holds for Qwen‑2.5‑1.5B/7B). Using the target set as control (i.e. no additional data) reproduces the reported trade-off, so DSAS's benefit is due to conditional gating, not to seeing more data than the baselines.
>
> **Broader Impact Concern.**
> We thank the reviewer for raising this point. We agree the dual-use risk is real, but note that it applies to activation steering as a family rather than to DSAS specifically. DSAS only governs when an existing steering map is applied and introduces no new capability for producing harmful content (which resides in the underlying steering maps, or more accessibly in base models via prompting or fine-tuning). We have made this explicit now in the Broader Impact Statement.

---

### Review · Reviewer_2r5p · 2026-07-05

**Summary Of Contributions:**

## Summary of Contributions

The manuscript proposes Dynamically Scaled Activation Steering (DSAS), a method to effectively introduce a local (layer-wise) gating mechanism to existing (LLM) steering techniques.
Such steering techniques guide the model away from harmful outputs in different ways, yet they don't necessarily adapt the *strength* of the guidance signal to the *necessity* to steer: even when outputs would not require steering, steering methods often still intervene, causing performance degradation.

This manuscript proposes a generic gating mechanism that modulates the strength of any steering method per token.
Small models are trained to classify activation of layers/blocks in the network as harmful or not, and modulate guidance strength correspondingly.
The manuscript then introduces a series of improvements on the practicality of this idea, such as averaging embeddings, PCA, accuracy-based deactivation of individual local gates, etc.

The proposed method is then validated on three language models and a modern text-to-image diffusion model, showing qualitative improvements in case studies and quantitative improvements in benchmarks.


## Strengths

The main strengths of the manuscript are the quality of writing, as well as the depth of the empirical evaluation and grounding in existing work:

### S1 Clear Motivation and Explanation of Setting and Idea

The manuscript is easy to follow, the setup of the problem and the basic idea as well importance of the approach become apparent quickly.
A central innovation on the setting of related methods is the introduction of a "control set", which should be unmodified.

### S2 Grounded in Existing Methods/Method Agnostic Approach

Since the approach is relatively generic, it can be applied post-hoc to existing methods.
The manuscript actively utilizes a variety of different steering techniques with their method to demonstrate that.

### S3 Careful Methodological and Practical Design

The manuscript builds a lot on the basic idea to improve the performance/stability of the method, such as strong dimensionality reduction, low-complexity classifiers, with focus on runtime implications and performance tradeoffs.

### S4 Thorough Empirical Validation

Many of the engineering decisions are supported by direct ablation studies.
This includes:
- Using PCA
- Scaling Strength $\lambda$
- Training set size
- deactivating low-accuracy classifiers or not

In this complicated setup, these ablation studies are important, as many of the parameter choices risk feeling arbitrary otherwise.

## Weaknesses
The weaknesses center around the limited conceptual justification for the specific design choices in the gating mechanism: it is not clear to me, from reading the manuscript, *why* the method should work like this, and *what* the objective will minimize in terms of the distributions of $\mathcal T,\mathcal S,\mathcal C$.
For example, I fear that simultaneous fitting of classifiers on activation distributions might lead to cascading distribution shifts, which is not theoretically investigated.
Furthermore, the manuscript becomes slightly vague with technical details at times.

### W1 Limited Conceptual Justification for Layer-wise Strength Modulation

From reading the manuscript, the idea of token-level intervention makes sense conceptually, and I understand that preexisting methods intervene layer/block-wise.
However in terms of the three datasets/their distributions, $\mathcal T,\mathcal S,\mathcal C$, it is unclear what precise objective is targeted.
This also extends to the linear interpolation and the (potentially uncalibrated?) confidence-based soft gating with a logistic regressor.
It is not necessary to provide full theoretical guarantees about the properties of the approach, but some intuition on why this method should be expected to work would help a lot.

### W2 Potential Issue: Distribution Shift caused by previous layers

The approach trains sequential gating mechanisms simultaneously on the unmodified activations in each layer.
Consequently, any distribution shift in later layers that is induced by actually using a steering method/the gating mechanisms in previous layers potentially distorts the distribution of activations enough that the later classifiers may become unreliable during inference.

Whether this actually happens in practice is unclear, but this might still be worth further investigation.

### W3 Section 3 mixes conceptual approach with implementation choices

Building on W1,W2 but potentially worth mentioning in isolation, Section 3 describes the approach of the paper directly as a pipeline with all engineering decisions.
It might be simpler to first clearly state that the approach aims to perform block-wise gating of the steering methods etc, and which problems will naturally arise, (e.g. the high dimensionality and low data regime etc.) and *afterwards* describe the pipeline.
This might be a subjective preference, but would make it easier to follow and assess the role of each design decision.

### W4 Table 1 and Figure 2: unclear threshold-based filtering may affect the comparison of methods

The caption of Table 1 reads
> Values are chosen to minimize ToxTET while
limiting PPLWik to at most a 5% increase and MMLU to at most a 3% decrease relative to the unmodified
model.

As written, this phrase is slightly ambiguous. If the following interpretation is correct, it affects how the results should be interpreted: the 5% and 3% thresholds may function as evaluation hyperparameters used to select the best $\lambda$ for each method. In that case, changing either threshold could potentially change the ranking of methods.
Furthermore, the paper does not appear to justify these particular thresholds, and the asymmetric tolerances for PPLWik and MMLU make the choice less self-evident. A clear justification, sensitivity analysis over these thresholds, or a more robust evaluation strategy such as reporting the full tradeoff curve would be helpful.

Also, while not detrimental, the reported numbers appear to lack confidence intervals or other indicators of statistical uncertainty/significance, which would make the comparison easier to interpret.

## Summary of Assessment

Overall, I find the paper well-motivated, clearly written, and empirically substantial. My main concerns are that some central design choices of the gating mechanism are not fully justified conceptually, and that some evaluation choices, especially the threshold-based filtering used for Table 1/Figure 2, require clarification to make the comparison easier to interpret.

**Additional Comments:**

Below a few notes on minor issues I found while reading.


- informal terminology in section 3.2: "src average" and "ctl average", same in alg. 1. line 11 and in the other algorithms "source"/"control" is often used instead of the introduced symbol $\mathcal S/\mathcal C$.
- typo in section 3.2 in the paragraph above **Dimensionality Reduction**: "beloning"
- unusual technical term below eq. 3.3: "the PCA and the **logistic**,..." this is unusual terminology (without "regression"), is this word used correctly?
- potential typo in page 22, Figure 8, "in this thesis"?
- there is some inconsistency in Algorithm 2 between lines 3/4 and lines 9/12, is $\hat \mathbf a$ meant in line 12?
- as presented, Fig 1. is slightly confusing to me, as the text input is at the bottom and the output at the top. swapping would make it easier for me to understand what is what.
- duplicate citation of Rodriguez et al (2025a,2025b)

**Audience:**

Yes

**Audience Explanation:**

Although I am not satisfied with the conceptual justification of the method, the empirical validation still indicates that it is useful in practice.
As such, I believe that the potential to outperform SOTA on an important problem such as mitigating harmful outputs warrants considering the submission interesting independently of my concerns.

Furthermore, the basic idea is general and elegant: constructing a simple gating mechanism can improve the performance of model intervention techniques such as steering.
Similar techniques might be in reach for adjacent problems in inference-time control/scaling.

**Broader Impact Concerns:**

Broader Impact Statement fine as present.

**Claims And Evidence:**

No

**Claims Explanation:**

I am torn on this question, but lean towards “no.”
Since the soundness of the approach is not fully clear conceptually (W1, W2), the support for the main claims is primarily empirical.
However, the empirical comparison to existing methods is difficult to interpret without further clarification of the threshold-based selection procedure in Table 1/Figure 2 (W4). In particular, if the reported results are sensitive to the 5% PPLWik and 3% MMLU constraints, then the evidence is not yet sufficiently reliable to support the claimed comparison to SOTA methods.

**Requested Changes:**

I would structure my change requests around my four stated weaknesses.
I would also be happy to clarify individual points through discussion with the authors.
The following list is sorted by decreasing severity.

## Critical for my recommendation:

### (W4) Clarify and justify the selection procedure in Table 1/Figure 2

As stated above, the caption of Table 1 states that
> Values are chosen to minimize ToxTET while
limiting PPLWik to at most a 5% increase and MMLU to at most a 3% decrease relative to the unmodified
model.

I would like the authors to clarify exactly how this selection was performed. In particular, were the 5% and 3% thresholds fixed independently of the reported results, and are they used to select the best $\lambda$ separately for each method?
What does a 3% decrease in MMLU mean: a relative decrease with respect to the initial value, or a decrease by 3 percentage points?
If this interpretation is correct, then I consider it important that the authors either justify these particular thresholds clearly or provide evidence that the conclusions are robust to reasonable alternative thresholds.
Ideally, the authors would use an evaluation protocol that does not depend on these constraint thresholds. Alternatively, they could provide a sensitivity analysis over the threshold values.

It is not necessary to report every datapoint, but the comparison should be presented such that the methods can be compared transparently and robustly.


###  (W1) Provide a clearer conceptual justification for the gating mechanism

The manuscript should more clearly explain why the proposed layer-wise/token-wise strength modulation should be expected to work. In particular, it would help to clarify what objective or distributional desideratum is implicitly targeted with respect to the target, steering, and control sets $\mathcal T,\mathcal S,\mathcal C$. I do not think full theoretical guarantees are necessary, but the paper should give more intuition for why classifier confidence, especially from potentially uncalibrated logistic regressors, is an appropriate quantity for continuously interpolating the steering strength.



## Important


### (W2) Address the possible cascading distribution shift induced by interventions in earlier layers

Since the gating classifiers are trained on unmodified activations, but are used during inference after earlier layers may already have been steered, there is a possible mismatch between the training and inference activation distributions for later-layer gates. I would like the authors to address this point either empirically or conceptually. For example, they could add a diagnostic showing that later-layer classifiers remain reliable under actual steered inference, an ablation using activations collected under steering, or a discussion explaining why this distribution shift is expected to be small in practice.

## Would strengthen the manuscript, but not critical

### (W3) Separate the conceptual method from implementation choices more clearly in Section 3

Section 3 currently presents the method largely as a complete pipeline, including several engineering choices. I think the paper would be easier to follow if it first stated the core conceptual idea more explicitly, namely block-wise/token-wise gating of steering strength, and then separately motivated the practical choices needed to make this feasible, such as averaging embeddings, PCA, low-complexity classifiers, and deactivation of low-accuracy gates. This is not critical for my recommendation, but it would make the method easier to understand and assess.

---

> ### Author Response · Authors · 2026-07-14
> **Rebuttal to Reviewer 2r5p**
>
> We appreciate the thorough review as well as recognizing the strengths of our paper. Next, we address the requested changes in importance order:
>
>
> **W4.**
> We thank the reviewer for this careful reading. We would like to clarify that Figure 2 is our main result and is threshold-free, and shows that methods with DSAS dominate the Pareto front. Table 1 is a single operating point within the Pareto front, which exists only to give a compact numeric side-by-side that also fits the conditional-steering baselines (MERA, CAST, AlphaSteer), which would clutter the Pareto plots if overlaid (Pareto plots for each conditional-steering baselines are found in Appendices M to O).
>
> We pick one comparable point per method via a fixed "acceptable-degradation" budget (≤5% PPL_Wik increase, ≤3% MMLU decrease). This budget is applied identically to every method and never tuned per method and it would not alter the relative ranking of DSAS with respect to other methods since it dominates the entire Pareto front.
>
> The difference between the two thresholds responds to the scale of the two quantities they bound. Perplexity is an exponential, higher-variance measure: a few percent relative change corresponds to only a small shift in cross-entropy loss (e.g., a move from 1.50 to 1.55 nats is already ≈+5% PPL) and is routinely within run-to-run noise, so a ~5% bound is a conservative "still fine" threshold for fluency. MMLU is a discrete accuracy over a large test set with sub-percent noise, where a ~3% drop is already near a conventional boundary for a clear capability regression. We have now added a sentence to the Table 1 caption clarifying that Figure 2 is our main threshold-free result.
>
> **W1.**
> The goal of DSAS is to steer the source distribution toward the target while leaving the control distribution unchanged (Section 3.2). One example where this is desirable is when the control set would be out of distribution with respect to the steering map.  We now make the role of the three sets explicit. The gate $h_\ell$ is trained on source vs. control and decides when to intervene, while the target set enters only through the steering map $T_\ell$, which decides where to map the source.
>
> The continuous coefficient targets a simple desideratum. We want to apply full steering to clearly source-like tokens, none to clearly control-like ones, and a graded blend in between (we have added this intuition to Section 3.2). The classifier posterior is the natural quantity for this, as it directly expresses graded source-vs-control membership. Continuous interpolation is then the smooth relaxation of a hard on/off gate. As the reviewer notes this relies on the confidence being meaningful, we verify empirically that the gate is well calibrated on held-out data (new Appendix J).
>
> **W2.**
> It is right that, in offline DSAS, each gating classifier is fit on unmodified activations, while at inference layer $\ell$ see activations already steered at layers $1,\dots,\ell-1$. We note however that every result in the paper is already measured under exactly this "shifted" regime, so all our results show that DSAS remains reliable under actual steered inference. DSAS detects toxic regions in activation space, so as earlier layers steer activations toward non-toxicity, later layers naturally apply weaker steering, avoiding over-steering.
>
> Additionally, in our end-to-end version, the classifiers are trained jointly for all layers, together with LinEAS maps. If the cascading shift degraded the offline gates, training under steering should heavily improve results, yet DSAS-E2E only matches or slightly outperforms vanilla DSAS (Fig. 2), showing the distinction makes little practical difference. We have now added clarifications in Sec. 3.2 and in Sec. 4.
>
> **W3.**
> We thank the reviewer for this helpful suggestion and agree it improves readability. We have restructured Section 3.2 to state the conceptual idea first and only then motivate the practical choices.
>
> Concretely, the subsection now opens with a Core idea paragraph that presents the central mechanism together with the interpolation equation (moved up from the end of the subsection) and the when vs. how decoupling. Only after this do we describe the engineering choices needed to make it feasible: training the linear gate, averaging embeddings, PCA regularization, and deactivating low-accuracy gates. This makes explicit which parts are conceptual and which are implementation, without changing any technical content.
>
> Finally, we want to thank the reviewer for the notes on minor issues. They have all been addressed in the manuscript.

---

### Review · Reviewer_1AtW · 2026-07-07

**Summary Of Contributions:**

The paper proposes Dynamically Scaled Activation Steering, a method-agnostic wrapper for activation steering. Existing activation-steering methods such as CAA, ITI, and LinEAS learn how to modify internal activations in order to push a model away from an undesired behavior and toward a desired behavior. However, these methods typically apply the intervention with a fixed global strength across inputs, layers, and tokens, which can degrade ordinary model behavior when steering is unnecessary. DSAS addresses this by learning when and how strongly to apply an existing steering transformation. For each layer, the method trains a logistic regressor to distinguish activations from a source set, representing undesired behavior, from a control set, representing behavior that should remain unchanged. The paper evaluates DSAS primarily on toxicity mitigation in language models, and show that DSAS improves the Pareto trade-off between toxicity reduction and utility preservation.


Strengths:
1. The paper identifies an important weakness of existing activation steering: unconditional steering is too blunt. A fixed steering strength can reduce undesirable behavior, but it may also distort benign generations. DSAS directly addresses this by decoupling how to steer from when to steer, which is a clean and useful conceptual framing.

2. The method is modular. DSAS can be placed on top of existing steering methods such as CAA, ITI, and LinEAS. This makes the contribution broadly applicable and easy to integrate into existing activation-steering pipelines.


Weaknesses:
1. One major weakness is that the paper relies on the logistic-regression output as a continuous steering coefficient, but does not adequately justify that this output is calibrated. The method treats h as a probability-like value: if the classifier outputs 0.8, the intervention is applied with 80% strength. However, the paper mainly reports classifier accuracy and downstream Pareto improvements. Accuracy does not imply calibration. A classifier can be accurate but still produce poorly calibrated probabilities. The paper should either provide calibration evidence, such as ECE or temperature-scaling experiments to provide stronger evidence.

2. Another weakness is the assumption that sentence-level supervision can support token-level localization. The authors explain that they average embeddings because individual token labels are unavailable. This is reasonable, but it creates a gap between training and inference. The heatmaps are suggestive, but the paper does not provide a rigorous quantitative localization benchmark. It is therefore hard to know whether DSAS is truly identifying toxic or concept-relevant tokens, or whether it is detecting broader context and spreading the intervention over correlated tokens.

3. The diffusion-model experiment is interesting but weaker than the language-model results. Blurring bananas, phones, etc is a toy proxy for concept suppression. It demonstrates modality generality, but it does not strongly establish that DSAS would work for realistic image-safety tasks.

**Audience:**

Yes

**Audience Explanation:**

The paper studies an adaptive steering method for deep learning and provides a simple, lightweight and modular design that has practical effects.

**Claims And Evidence:**

No

**Claims Explanation:**

The weakness points 1 and 2 are not very well justified.

**Requested Changes:**

Please address the weaknesses above.

---

> ### Author Response · Authors · 2026-07-14
> **Rebuttal to Reviewer 1AtW**
>
> We appreciate the author the feedback. Following, we reply to the main concerns:
>
> **W1.** We agree that accuracy does not imply calibration, and we have added the requested evidence (new Appendix J). We evaluate the deployed gate (PCA-5 + logistic regression, trained on the 32/32 source/control sentences) on a larger held-out set of 600 labeled toxicity comments disjoint from the training data, and report ECE with and without temperature scaling (Guo et al., 2017):
>
> |Model | Acc. | AUROC | ECE | ECE (temp.-scaled) |
> |---|---|---|---|---|
> |Qwen2.5-1.5B | 0.76 | 0.84 | 0.025 | 0.027 |
> |Gemma-2-2B | 0.78 | 0.86 | 0.032 | 0.026 |
> |Qwen3-14B | 0.80 | 0.88 | 0.049 | 0.036 |
>
> The reliability diagrams are near-diagonal and temperature scaling changes ECE by at most $\approx 0.01$. This means  that on the held-out data the gate is already well calibrated, with no systematic miscalibration for temperature scaling to correct. We thank the reviewer for prompting this analysis, which we believe strengthens the paper.
>
> **W2.**
> To move our localization evidence beyond the qualitative heatmaps, we have added a quantitative localization benchmark (new subsection in Appendix I) that directly settles the question. We score the frozen, sentence-supervised gate per token against human token-level annotations in Toxic Spans Detection (Pavlopoulos et al., 2021). To separate localization from detection, we report a within-toxic metric that ranks annotated toxic tokens against other tokens inside the same toxic posts — a model that merely senses global toxic context would score at chance here. Across all three models the gate ranks the human-marked toxic tokens well above chance (within-toxic AUROC 0.70–0.76; per-post mAP >3× chance), quantitatively demonstrating that it concentrates on the toxic tokens rather than spreading over correlated context.  We stress, however, that exact agreement with the human spans is not the objective. Toxicity is represented contextually, so the gate can legitimately modulate contextual tokens beyond the annotated words. Its role is to make steering selective rather than uniform, and the clear above-chance separation we report already achieves this, which is what yields the Pareto gains (Section 4.1).
>
> **W3.**
> We agree the blurring proxy doesn't establish performance on real image-safety tasks, but that isn't its role. Contribution (v) is a generality claim that the when/how decoupling transfers to a non-autoregressive, spatial generator without modification. The evidence for suppressing genuinely unsafe behavior comes from the text experiments, which evaluate three LLMs, three steering methods, and three SOTA conditional baselines on real toxicity using an external classifier rather than a proxy.
>
> The use of placeholder concepts is deliberate (Section 4.2) because we avoid generating or displaying unsafe imagery. Within that constraint, our six concepts span meaningfully different visual structures, from small localized objects (bananas, phones) to full-scene compositions (castles, elephants), and DSAS shows the same selective-intervention behavior across all of them (Fig. 3–4, Appendix Q.2). Since the gate conditions only on activation separability rather than concept semantics, this structural diversity is what generalization should be judged on rather than the concept label.
>
> We will make this scoping explicit in the revision. The diffusion result supports modality generality, while the safety-relevant evidence should be interpreted from the text experiments.